# Aggregating forecasts of multiple respiratory pathogens supports more accurate forecasting of influenza-like illness

**Sen Pei** *, **Jeffrey Shaman** *

Department of Environmental Health Sciences, Mailman School of Public Health, Columbia University, New York, NY, United States of America

* sp3449@cumc.columbia.edu (SP); jls106@cumc.columbia.edu (JS)

## Abstract

Influenza-like illness (ILI) is a commonly measured syndromic signal representative of a range of acute respiratory infections. Reliable forecasts of ILI can support better preparation for patient surges in healthcare systems. Although ILI is an amalgamation of multiple pathogens with variable seasonal phasing and attack rates, most existing process-based forecasting systems treat ILI as a single infectious agent. Here, using ILI records and virologic surveillance data, we show that ILI signal can be disaggregated into distinct viral components. We generate separate predictions for six contributing pathogens (influenza A/H1, A/H3, B, respiratory syncytial virus, and human parainfluenza virus types 1–2 and 3), and develop a method to forecast ILI by aggregating these predictions. The relative contribution of each pathogen to the total ILI signal is estimated using a Markov Chain Monte Carlo (MCMC) method upon forecast aggregation. We find highly variable overall contributions from influenza type A viruses across seasons, but relatively stable contributions for the other pathogens. Using historical data from 1997 to 2014 at US national and regional levels, the proposed forecasting system generates improved predictions of both seasonal and near-term targets relative to a baseline method that simulates ILI as a single pathogen. The hierarchical forecasting system can generate predictions for each viral component, as well as infer and predict their contributions to ILI, which may additionally help physicians determine the etiological causes of ILI in clinical settings.

## Author summary

Influenza-like illness (ILI) is a widely used medical diagnosis of possible infection with influenza or another acute respiratory illness. Accurate forecasting of ILI can support better planning of interventions against respiratory infections, as well as early preparation for patient surges in healthcare facilities during periods of peak incidence. Although ILI is an amalgamation of multiple pathogens with variable seasonal phasing and contributions to incidence, to our knowledge, all existing process-based forecasting systems treat ILI as a single infectious agent. This leads to model misspecification that compromises forecast precision. In this study, we address this issue by forecasting ILI as the aggregation of

**Data Availability Statement:** Data and code are available at GitHub https://github.com/SenPei-CU/Multi-Pathogen_ILI_Forecast.

**Funding:** This work was supported by US National Institutes of Health grant GM110748 and Defense

Advanced Research Projects Agency contract
W911NF-16-2-0035. The funders had no role in the
study design, analysis, decision to publish, or
preparation of the manuscript.

**Competing interests:** JS and Columbia University
declare partial ownership of SK Analytics. JS
discloses consulting for BNI and Merck. SP
declares no competing interests.

predictions for individual contributing respiratory viruses. Using ILI records and viro-
logic surveillance data, we show that ILI signal can be disaggregated into distinct viral
components and develop a method to forecast ILI by aggregating predictions for six path-
ogens. We find highly variable overall contributions from influenza type A viruses across
seasons, but relatively stable contributions from other pathogens. In retrospective fore-
casts, the proposed multi-pathogen forecasting system generates substantially more accu-
rate predictions of both seasonal and near-term targets relative to a baseline method that
simulates ILI as a single pathogen.

## Introduction

Acute respiratory infections impose heavy morbidity and mortality burdens on global popula-
tions, especially children and the elderly [1]. Influenza-like illness (ILI), defined as fever (tem-
perature of 100˚F [37.8˚C] or greater) with a cough and/or sore throat, is a widely used
medical diagnosis of possible infection with influenza or another acute respiratory illness. The
US Centers for Disease Control and Prevention (CDC) collects surveillance information on
the percentage of patient visits to healthcare providers for ILI through the US Outpatient Influ-
enza-like Illness Surveillance Network (ILINet) and has adopted ILI as the primary signal to
track influenza activity in the United States [2]. Accurate forecasting of ILI can support better
planning of both pharmaceutical (e.g., vaccines, antivirals and prophylaxis) and non-pharma-
ceutical (e.g., school closure, social distancing and travel restrictions) interventions against
respiratory infections, as well as early preparation for patient surges in healthcare facilities dur-
ing peak periods of incidence. In recent years, a number of forecasting systems for ILI have
been developed [3–14], many of which have been applied operationally to forecast ILI in the
United States [15–18].

Despite its name, ILI is not exclusively caused by influenza. Common viruses contributing
to ILI include influenza, respiratory syncytial virus, human parainfluenza virus, coronavirus,
human metapneumovirus, respiratory adenovirus and rhinovirus [19–22]. Typically clinically
indistinguishable, these viral signals have disparate seasonal characteristics, and vary in their
contributions to ILI during different times of the year. Although ILI is a syndromic record that
represents a range of illnesses, to our knowledge, all current process-based forecasting systems
treat ILI as a single pathogen [16–18]. Such oversimplification may lead to model misspecifica-
tion that compromises forecast precision. In addition, single-pathogen forecasting systems are
unable to estimate and predict the relative contribution of each component pathogen.

To generate more accurate and precise ILI forecasts, the inclusion of more complex pro-
cesses is needed to reduce model misspecification. Currently, laboratory-confirmed positivity
rate data (defined as the fraction of samples testing positive for a specific respiratory pathogen)
for several respiratory pathogens circulating in the US are available in near-real time from the
National Respiratory and Enteric Virus Surveillance System (NREVSS) [23]. These pathogen-
specific observational data might support prediction of each virus singly, and further improve
the precision of ILI forecasting. Indeed, previous research has found that forecasting individual
pathogens produces more precise predictions of those specific agents due to better-specified
models [24–26].

Here, using ILI records from ILINet and virologic surveillance data from the NREVSS, we
show that ILI signal can be recovered by aggregating multiple viral components, specifically
influenza A/H1, A/H3, B, respiratory syncytial virus (RSV), and human parainfluenza virus
types 1–2 (PIV12) and 3 (PIV3). Further, we demonstrate that more accurate and precise

forecasts of ILI can be obtained by aggregating predictions of these pathogens. Retrospective forecasts for the 1997–1998 to 2013–2014 seasons at national and regional levels demonstrate that this multi-pathogen forecasting system generates substantially improved predictions of both seasonal (onset week, peak week and peak intensity) and near-term (one- to four-week ahead ILI) targets relative to a competing method modeling ILI as a single agent. The multi-pathogen forecasting system, capable of generating predictions for each virus, can infer and predict their relative contributions to ILI as well.

## Materials and methods

### Data

Weekly ILI records for US national and 10 HHS (US Department of Health and Human Services) regions were obtained from the CDC FluView website [2]. Positivity rate data for influenza types and subtypes are also available from FluView. In this study, we focus on three influenza types/subtypes–A/H1, A/H3 and B. Weekly laboratory test results for RSV, PIV12 and PIV3 at national and regional levels were obtained from NREVSS. Positivity rate data for several other respiratory pathogens (e.g., coronavirus, human metapneumovirus and respiratory adenovirus) are also reported but with limited historical records. We therefore constrained study to the six viruses with data since 1997 –influenza A/H1, A/H3, B, RSV, PIV12 and PIV3. Participating laboratories voluntarily report to the NREVSS the total number of weekly tests performed to detect each virus and the total number of positive tests (including virus antigen detections, isolations by culture, and polymerase chain reaction (PCR)). The participating laboratories are spread throughout the US [23] and are thus fairly geographically representative of the broader population. However, as testing was ordered independently by healthcare providers, it is unknown whether samples were biased to certain group of patients.

In total, we used ILI and laboratory test data from the 1997–1998 to 2013–2014 seasons, excluding 2008–2009 and 2009–2010, which were impacted by the 2009 H1N1 pandemic (Fig 1,

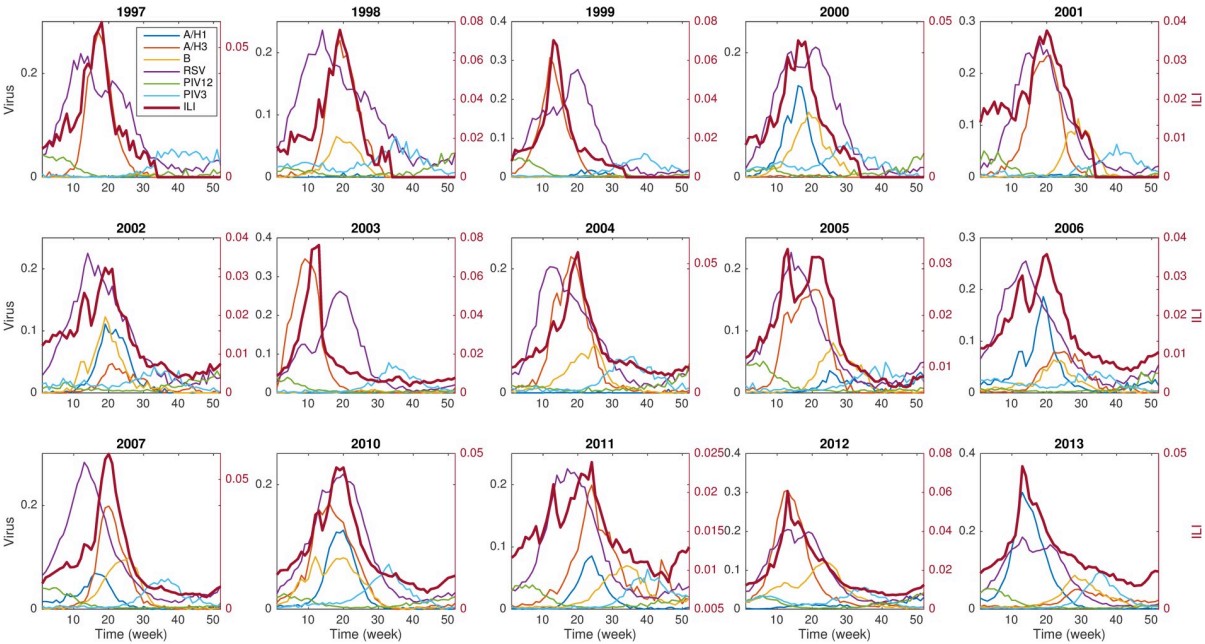

**Fig 1. National ILI rates and six contributing viral signals in different seasons.** The viral signal shown here is the positivity rate of each pathogen. ILI is attributed to a range of viruses with variable seasonal phasing. ILI peaks are largely driven by the circulating influenza strains.

S1 and S2 Figs). Retrospective forecasts were generated at the US national level and for HHS regions 1 to 9. Region 10 was excluded as influenza positivity data were only available after 2010.

## Forecasting single respiratory pathogens

The trajectories of RSV, PIV12 and PIV3 are similar across seasons, whereas the activity of each influenza strain is highly variable (S1 Fig). As a result, we adopted two distinct approaches to forecast influenza and non-influenza pathogens. For influenza, we used process-based models that could flexibly simulate a variety of outbreak trajectories. A humidity-forced SIRS (susceptible-infected-recovered-susceptible) model was employed to depict the influenza transmission process and iteratively optimized using available observations. In particular, we fitted an ensemble of model simulations to positivity rate data from the beginning of the outbreak season up to the week of forecast [27–29] and then evolved the simulations into the future to generate forecasts. We also utilized a method to optimally perturb the fitted model to obtain a more calibrated ensemble forecast spread [30]. The spread of forecasts was calibrated to capture the uncertainty of near-term observations. Specifically, the initial conditions (S and I) of the fitted model were perturbed so that the difference between the spread of near-term model trajectories and observations is minimized (see details in S1 Text). In implementation, a 1,000-member ensemble forecast was generated each week.

For the non-influenza pathogens with consistent seasonality, we generated forecasts using a statistical approach that leverages the similarity among different seasons–the method of analogues [14]. Several observational studies indicate that both RSV and PIV have relatively regular seasonality in temperate regions [31,32]. In the US, RSV typically peaks in early February, and PIV3 and PIV2 usually peak in April-June and October-November, respectively [33,34]. The seasonality of RSV has been found correlated with climate factors such as temperature and humidity [35,36]. However, the exact drivers of the seasonality of RSV and other respiratory viruses and their comparative significance are still under debate. For the method of analogues, we measured the distance of the time series of observations in the current season to those in other seasons in the same location, and produced forecasts as a weighted average of historical records based on those distances. To generate probabilistic predictions, 1,000 time series were sampled from historical records with replacement according to their weights, and then calibrated to optimize the forecast spread. Details about the forecasting of influenza and non-influenza pathogens are presented in S1 Text.

## Aggregating predictions of multiple pathogens

To generate ILI forecasts by aggregating predictions of the contributing viral signals, we estimated a multiplicative factor for each pathogen. Specifically, using the ILI data and positivity rates for the 6 viruses, we employed a Markov Chain Monte Carlo (MCMC) method to infer the multiplicative factors $w_i$ [37]. To ensure a broad probing of $w_i$, we used a uniform prior distribution $U(0,1)$ in the MCMC. Distributions of six multiplicative factors were obtained to maximize the log likelihood of observing the reported ILI data (see S1 Text for details). In implementation, the distributions of multiplicative factors were re-estimated each forecast week. In order to generate an ensemble of ILI forecast trajectories, 1,000 samples of multiplicative factors were randomly drawn from the posterior distributions, and then used to aggregate the previously obtained 1,000-member ensemble forecast for each pathogen. In the aggregation, multiplicative factors and forecast trajectories were randomly matched.

A post-processing procedure was applied to further improve the accuracy and precision of the ILI forecasts [6,38]. Systematic bias across seasons may exist in ILINet reported

surveillance data, such as the drops in ILI often observed during the Christmas-New Year week. We accounted for this systematic bias by adding a component estimated from the residuals of ILI fits from other seasons (S3 Fig). After adjusting for the bias common to all seasons, real-time forecasts may still under or overestimate the ILI trajectory for a particular prediction. We therefore included another adjustment to counteract the bias specific to each forecast [6,38], which was approximated as the discrepancy between the aggregated and observed ILI data in the latest week. Finally, the forecast ensemble was redistributed around the mean prediction to calibrate its spread. Implementation details about the MCMC and post-processing are provided in S1 Text.

We acknowledge that ILI is attributable to more viruses than just the 6 pathogens included here; for instance, coronavirus, human metapneumovirus, respiratory adenovirus and rhinovirus may all produce symptoms consistent with an ILI diagnosis. However, without pathogen-specific data for those viruses, it is difficult to estimate the relative contribution of such neglected pathogens. For the 2010–2011 through 2013–2014 seasons, positivity rate data for three additional pathogens were available: human metapneumovirus (HMPV), respiratory adenovirus (rAD) and rhinovirus (Rhino). We therefore performed an additional analysis estimating the relative contributions of nine pathogens to ILI, and found that, at national level, the three additional pathogens accounted for 21%, 37%, 19% and 31% of the ILI signal in those four seasons (S4 Fig). This finding indicates that the principal six pathogens considered here comprise the majority of ILI. In aggregation, the contributions of neglected pathogens are partially represented as forecast discrepancy or by viruses with similar seasonality. In future studies, with the availability of more abundant pathogen-specific data, we will include more viruses in the multi-pathogen aggregation.

## Evaluation of forecasts

The accuracy of probabilistic forecasts was measured using the "log score", which is calculated as the logarithm (base $e$) of the fraction of forecasts assigned to an interval around the observed target (henceforth, score interval) [16–18]. We focus on four near-term targets (1- to 4-week ahead ILI) and three seasonal targets (peak week, peak intensity and onset week) [16–18]. In particular, peak week was defined as the week with the highest ILI rate in each season, and onset was defined as the first of three consecutive weeks with ILI at or over a predefined threshold. These thresholds for the US and all HHS regions were released by CDC prior to each season. For 1- to 4-week ahead ILI and peak intensity, the score interval was set as ±0.5% around the observed ILI rate; for peak week and onset week, it was set as ±1 week around the observation. A floor value of -10 was set as a lower bound for scores. Translating log score to forecast skill, a -0.5 (-1.0) log score implies a correct forecast approximately 61% (37%) of the time. Another measure, forecast error, calculated as the mean absolute error of average predictions to observed targets, was used to quantify the accuracy of point predictions. These forecast metrics are consistent with CDC FluSight guidelines [16–18]. Note that, starting from the 2019–2020 season, an updated scoring rule using a single data bin was employed [39]. In this study, we used the previous definition for log score.

## Results

### ILI disaggregation

The six examined pathogens have distinct outbreak characteristics (Fig 1, S1 Fig). Influenza viruses typically peak during winter but exhibit large variations in peak week and peak intensity across seasons. This irregular behavior renders influenza forecasting a challenging task. In contrast, the activity of RSV, PIV12 and PIV3 is more regular: these viruses generally peak in

January, November and May, respectively, with similar peak intensity across seasons. ILI epidemic trajectories are highly variable, with peak timing largely driven by the circulating influenza strain in each season. We show the scatter plot for each possible pair of observations (S2 Fig), and found that, in general, influenza and RSV are positively correlated with each other, and PIV12 and PIV3 are negatively correlated with influenza and RSV. In addition, PIV12 is negatively correlated with PIV3.

We represent ILI signal in a given season $s$ as a linear combination of the positivity rates of its contributing pathogens in the same season (S1 Text): $ILI(t) = \sum w_i(t)v_i(t)$, where $v_i(t)$ is the positivity rate of pathogen $i$ at week $t$, and $w_i(t)$ is its corresponding multiplicative factor. Here we omit the season index $s$ for notational convenience. We derived that the multiplicative factor $w_i(t)$ quantifies the ratio of the probability of undergoing a laboratory test among patients seeking medical attention for any reason to the probability of undergoing a laboratory test among patients seeking medical attention who are infected with pathogen $i$ (S1 Text). As a result, the multiplicative factors for the six pathogens do not necessarily sum to 1. If a multiplicative factor $w_i(t)$ is stable over time, i.e., $w_i(t) = w_i$, we can predict ILI by aggregating forecasts for all contributing pathogens. To validate this assumption, we performed a linear regression to ILI with a constant multiplicative factor, $w_i$, for each pathogen.

We fitted the positivity rate data of each virus to a mechanistic model (humidity-forced SIRS models for influenza A/H1, A/H3 and B; SIRS models for RSV, PIV12 and PIV3) and used the fitted curves to represent the actual unobserved positivity rates (Fig 2A). We then performed a linear regression to ILI using the fitted curves via MCMC in order to derive estimates for $w_i$ (Fig 2B). For comparison, we also fitted the ILI data to a humidity-forced SIRS model with ILI as a single pathogen.

The mechanistic models simulated the positivity rates of each pathogen well (Fig 2A), indicating that the process-based model is well specified for the transmission process of a single virus. This finding is also evident from the improved forecast accuracy for specific influenza types and subtypes reported in prior work [25]. The epidemic curve aggregated from the six pathogens closely matches the observed ILI data (Fig 2C). (Note that each component in the multi-pathogen aggregation needs to fit the observed positivity rate of each pathogen as well.) In contrast, the curve fitted to ILI as a single pathogen has discrepancies from observations during early and late weeks. These findings are corroborated across all seasons and regions (S1 Table, S5 Fig) and indicate that the multi-pathogen model with constant multiplicative factors is better specified for simulation of ILI. The distributions of multiplicative factors inferred for each season differ substantially, indicating that the contributions of the individual pathogens to overall ILI vary (S6 Fig). Within a season, distributions of multiplicative factors in different regions also have variations; however, the relative magnitudes for influenza types/subtypes remain similar across regions (S7 and S8 Figs). Note, the fitting did not reproduce the decrease of ILI at week 14 during the Christmas-New Year holiday. This discrepancy will be modeled during forecast post processing as an independent component representing systematic bias.

The overall contributions of the six pathogens to ILI in 15 seasons at the national level are reported in Table 1. The estimated contributions are relatively stable for non-influenza pathogens and influenza type B, but exhibit large year-to-year variability for influenza viruses A/H1 and A/H3: the standard deviations for A/H1, A/H3, B, RSV, PIV12 and PIV3 are 13.4%, 20.2%, 6.8%, 5.7%, 8.1% and 2.8%, respectively. The variance of contribution from influenza B is similar to non-influenza pathogens. However, the seasonality of influenza B is more irregular. This highlights the variable prevalence of influenza type A across seasons owing to antigenic drift and immune escape. Each season also tends to have only one dominant influenza type A virus (A/H1 or A/H3), leaving the other sub-type circulating with much lower activity.

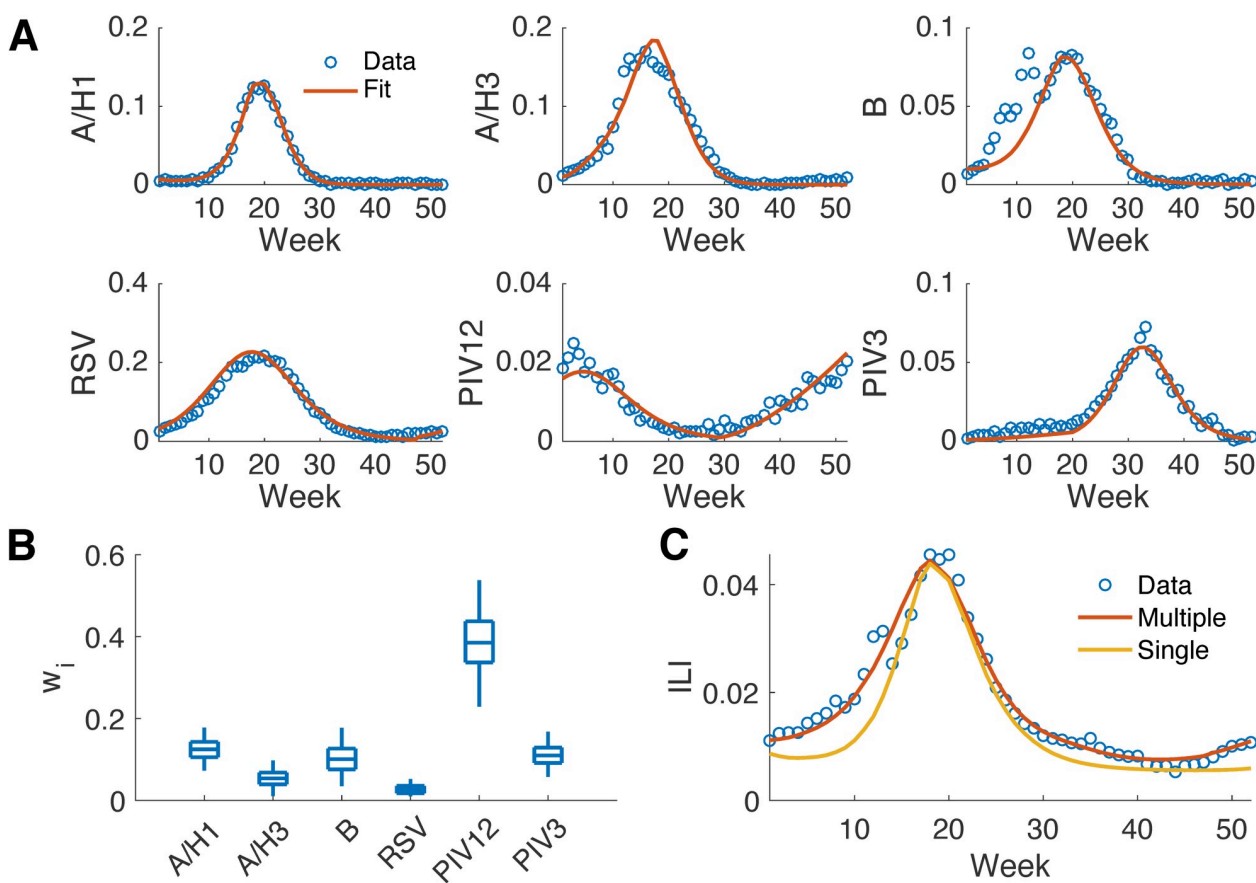

**Fig 2. Disaggregation of national ILI during the 2010–2011 season to multiple viral components.** Model fitting to the positivity rates of six respiratory pathogens (A). Time starts from the 40[th] week in 2010. The distribution of the multiplicative factor, $w_i$, for each pathogen obtained from MCMC (B). The boxes and whiskers show the interquartile and 95% credible intervals. Fitting ILI as an amalgamation of six pathogens versus as a single agent (C).

The different seasonality between influenza and other examined respiratory pathogens can be attributed to a number of factors. For instance, influenza transmission dynamics depend on pre-exposure to influenza viruses and antigenic drift, which modulate rates of susceptibility [40–42]. RSV mostly infects infants or young children whose maternal immunity has waned [43]. The complex interaction between different influenza strains and other respiratory viruses may also contribute to the variations in different seasons [44–46]. In addition, a recent study reported that the virology and host response to RSV and influenza are distinct. While influenza induces robust strain-specific immunity after infection, strain-specific protection for RSV is incomplete [47].

### Retrospective forecasting of ILI

Predictions for influenza A/H1, A/H3 and B generated by process-based models capture observed trajectories well (see examples in S9 Fig). For the non-influenza pathogens, RSV, PIV12 and PIV3, forecasts generated using the method of analogues reliably predict the positivity rates as well (S9 Fig). We also find that the prediction intervals generated for near-term targets are well-calibrated with the spread of observations (S10 Fig), i.e., 50% of observed outcomes fall within the interquartile prediction interval; 95% of observed outcomes fall within the 95% prediction interval; etc.

**Table 1. Overall contributions of the six pathogens to national ILI in 15 seasons.** The contribution of pathogen $i$ is calculated as $\sum_{t=1}^{52} w_i v_i(t) / \sum_{i=1}^{6} \sum_{t=1}^{52} w_i v_i(t)$, where $v_i(t)$ is the observed positivity rate of pathogen $i$ at week $t$, and $w_i$ is the inferred mean multiplicative factor for pathogen $i$. The inferred dominant pathogen in each season is in bold.

| | A/H1 | A/H3 | B | RSV | PIV12 | PIV3 |
|---|---|---|---|---|---|---|
| 97–98 | 0.00% | **43.10%** | 0.50% | 31.20% | 17.70% | 7.50% |
| 98–99 | 0.30% | **35.40%** | 12.80% | 28.80% | 15.70% | 7.00% |
| 99–00 | 1.90% | **60.40%** | 0.50% | 18.60% | 8.80% | 9.80% |
| 00–01 | **30.00%** | 0.20% | 15.10% | 21.40% | 24.40% | 8.90% |
| 01–02 | 0.50% | **34.40%** | 8.50% | 26.40% | 24.40% | 5.80% |
| 02–03 | 8.50% | 2.60% | 14.90% | **31.20%** | 29.60% | 13.20% |
| 03–04 | 0.00% | **65.80%** | 1.60% | 19.10% | 2.20% | 11.30% |
| 04–05 | 0.10% | **35.80%** | 14.50% | 15.80% | 21.00% | 12.80% |
| 05–06 | 3.70% | **31.30%** | 12.70% | 22.00% | 18.60% | 11.70% |
| 06–07 | **27.60%** | 5.40% | 9.20% | 25.60% | 15.70% | 16.50% |
| 07–08 | 12.20% | **34.10%** | 12.80% | 19.70% | 12.20% | 9.00% |
| 10–11 | 18.10% | 15.60% | **20.20%** | 15.00% | 18.40% | 12.70% |
| 11–12 | 5.50% | 18.10% | 8.30% | **28.90%** | 28.40% | 10.80% |
| 12–13 | 3.60% | **39.10%** | 25.30% | 15.50% | 6.50% | 10.00% |
| 13–14 | **42.80%** | 5.10% | 15.80% | 17.20% | 10.50% | 8.60% |

Fig 3 shows an example multi-pathogen forecast generated at week 6 of the 2010–2011 season at the national level. The aggregated forecast for ILI, improved by the post-processing that adjusts systematic bias, forecast-specific bias and forecast spread (Fig 3A), agrees well with the observed data. The adjustment for systematic bias reproduces the decrease of ILI in week 14; correcting forecast-specific bias reduces the discrepancy between predicted and observed ILI; and the calibration constrains the prediction intervals more tightly around the observed ILI curve. Near-term targets in the next one to four weeks fall within the 95% prediction intervals. The distributions of multiplicative factors estimated at week 6 (Fig 3B) are similar to those obtained from linear regression using data for the entire season (Fig 2B), but with a broader spread indicating a higher level of uncertainty. In comparison, the forecast obtained using a method that simulates ILI as a single pathogen can generally predict the trend of ILI but is less precise (Fig 3C). More forecast examples at different outbreak phases indicate that the aggregation of forecasts can well predict future ILI rates throughout an outbreak season (S11, S12, S13 and S14 Figs).

To evaluate the proposed ILI forecasting method, we performed retrospective forecasts for ILI in the US and HHS regions 1 to 9 during 15 seasons, using both the multi-pathogen approach and a baseline method that models ILI as a single agent [3]. To avoid over-fitting and to make full use of surveillance data, we used a "leave-one-out" cross-validation. That is, we trained the forecast algorithm using data from the periods outside the forecast season, so that the prediction is not contaminated by observations in the same outbreak. Within each season, weekly forecasts from week 4 to week 35 (i.e., late October to late May) were generated. Comparisons of log scores for seven targets averaged over forecasts in different seasons and locations indicate that the aggregated forecasts have significantly better log scores than the baseline forecasts (Wilcoxon signed-rank test, $p < 10^{-5}$, see details in S1 Text, Fig 4A, S15 Fig). Without post-processing, the aggregated forecasts already outperform the baseline; each of the post-processing procedures leads to additional improvement of the aggregated forecasts (S16 Fig). Specifically, improvement of the seasonal targets is primarily attributed to the forecast aggregation, and the adjustment for forecast-specific bias substantially improves near-term

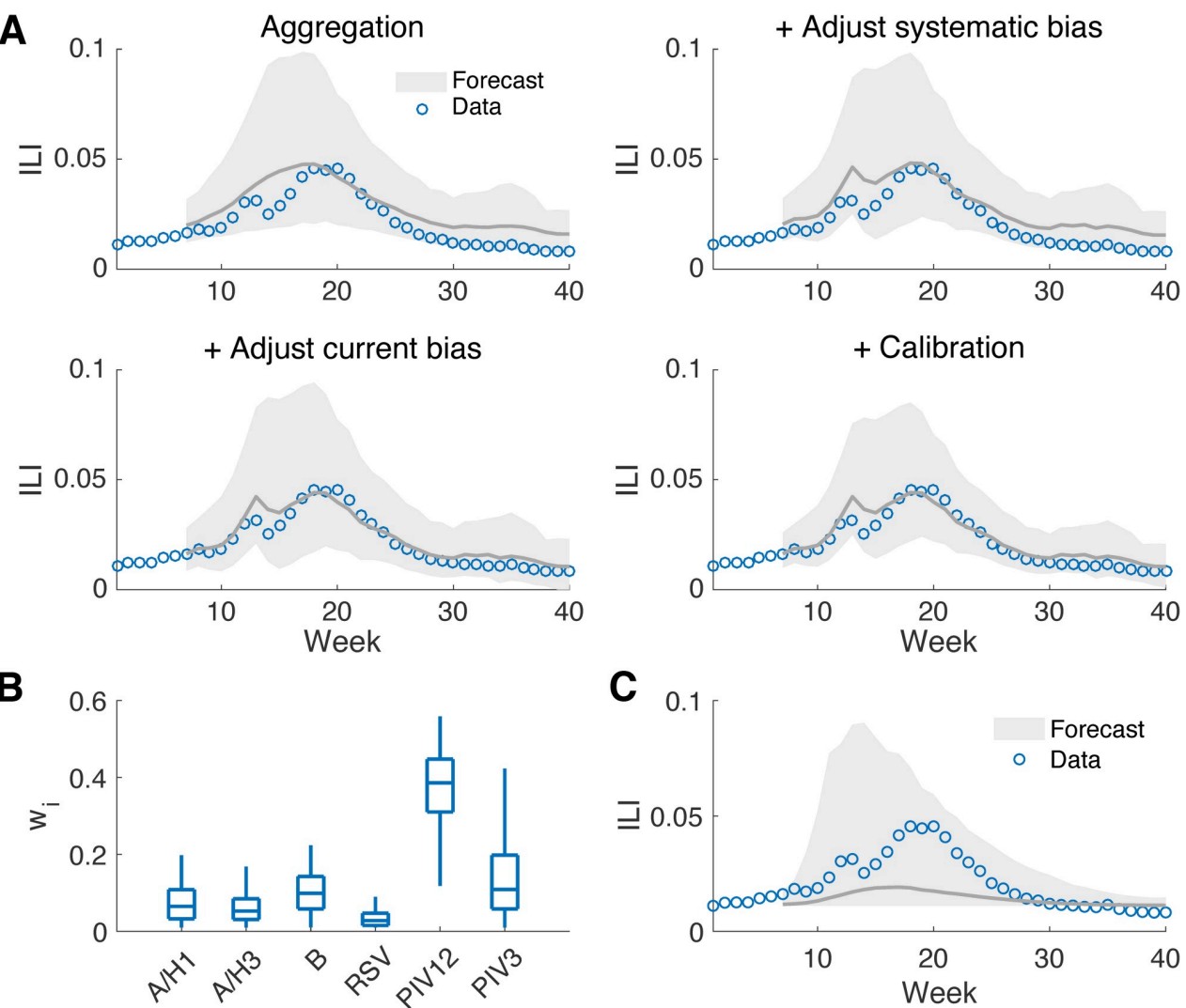

**Fig 3. Forecasts generated for national ILI during the 2010–2011 season by aggregating multiple viral components.** Forecast of ILI generated by aggregating the predictions for the six viruses using MCMC-derived multiplicative factors and the effect of post-processing (A). Forecasts were generated at week 6 in the 2010–2011 season. The grey line shows the mean trajectory and the shaded area indicates the 95% prediction interval. The distributions of the inferred multiplicative factors for the six pathogens at week 6 are reported in (B). The boxes and whiskers show the interquartile and 95% credible intervals. The baseline forecast treating ILI as a single pathogen is shown in (C).

targets. The benefit of bias correction post-processing was recently demonstrated for ILI forecasting [6]. Combining both forecast aggregation and post-processing, the log scores for 1- to 4-week ahead ILI, peak week, peak intensity, and onset week are improved by 0.38, 0.36, 0.33, 0.32, 0.18, 0.27 and 0.15, respectively. Comparisons based on forecast error for point predictions (Fig 4B and 4C) also demonstrate superior performance of the multi-pathogen forecast for most targets (except for 4-week ahead prediction); the forecast errors for 1- to 3-week ahead ILI, peak week, peak intensity, and onset week are reduced by 31.9%, 16.8%, 3.3%, 22.9%, 16.8% and 56.4%. By both evaluation metrics, the improvement for near-term targets gradually decreases for longer forecast horizons.

We evaluate forecast precision using reliability plot. For the probabilistic forecasts, the prediction intervals generated by the forecasting system should be well calibrated with the spread of observations (i.e. 50% of observed outcomes fall within the interquartile prediction interval;

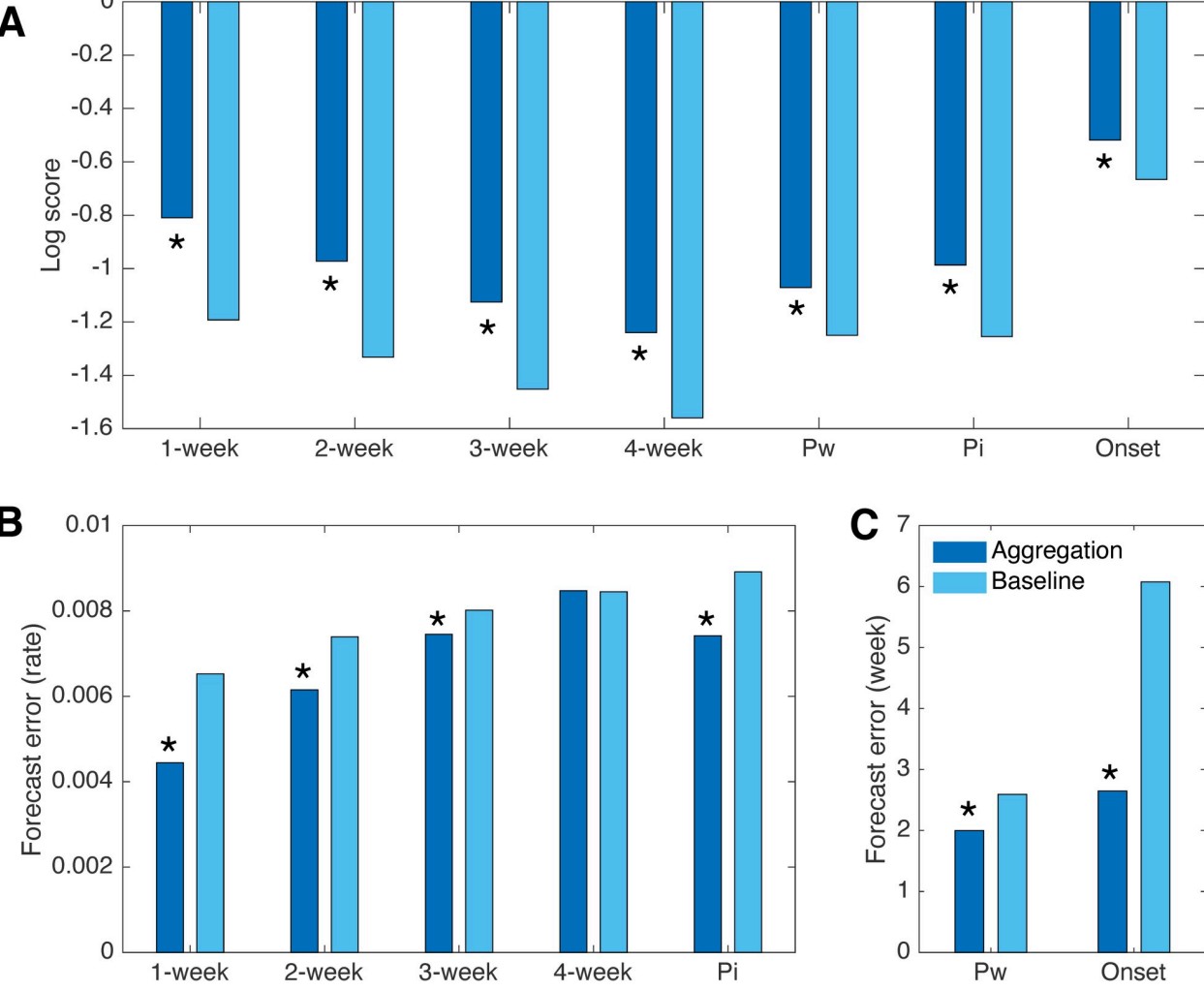

**Fig 4. Comparison of forecast performance with the baseline method.** Comparisons of log scores for forecasts generated by aggregating multiple pathogens and the baseline method that simulates ILI as a single agent (A). Log scores are obtained from weekly retrospective forecasts for ILI at the national level and in HHS regions 1 to 9, from 1997–1998 to 2013–2014 seasons, excluding pandemic seasons 2008–2009 and 2009–2010. Four near-term targets (1- to 4-week ahead ILI) and three seasonal targets (peak week, peak intensity and onset week, denoted by Pw, Pi and Onset, respectively) are shown. Comparison of forecast error for predictions of 1- to 4-week ahead ILI and peak intensity (B). Comparison of forecast error for predictions of peak week and onset week (C). Asterisks indicate statistical significance for multi-pathogen forecasts outperforming baseline ($p<10^{-5}$, Wilcoxon signed-rank test).

95% of observed outcomes fall within the 95% prediction interval; etc.). We evaluated this property using reliability plots. Specifically, we calculated the fraction of observed targets falling within the 25%, 50%, 75% and 100% prediction intervals, and display the relationship between the observed fractions and prediction intervals in Fig 5. A well-calibrated forecast yields data points that lie on the diagonal $y = x$ line. Results in Fig 5 indicate that the calibration for near-term targets and peak intensity is substantially improved by the aggregated forecasting approach. In contrast, the baseline method tends to generate overly narrow prediction intervals (data points lie below the diagonal line). Indeed, the average width of 95% prediction intervals generated by the baseline is narrower than those generated by the aggregation method (S17 Fig).

We also found that using the systematic bias correction procedure alone with the baseline method does not provide clear improvement of its performance (S18 Fig). We additionally

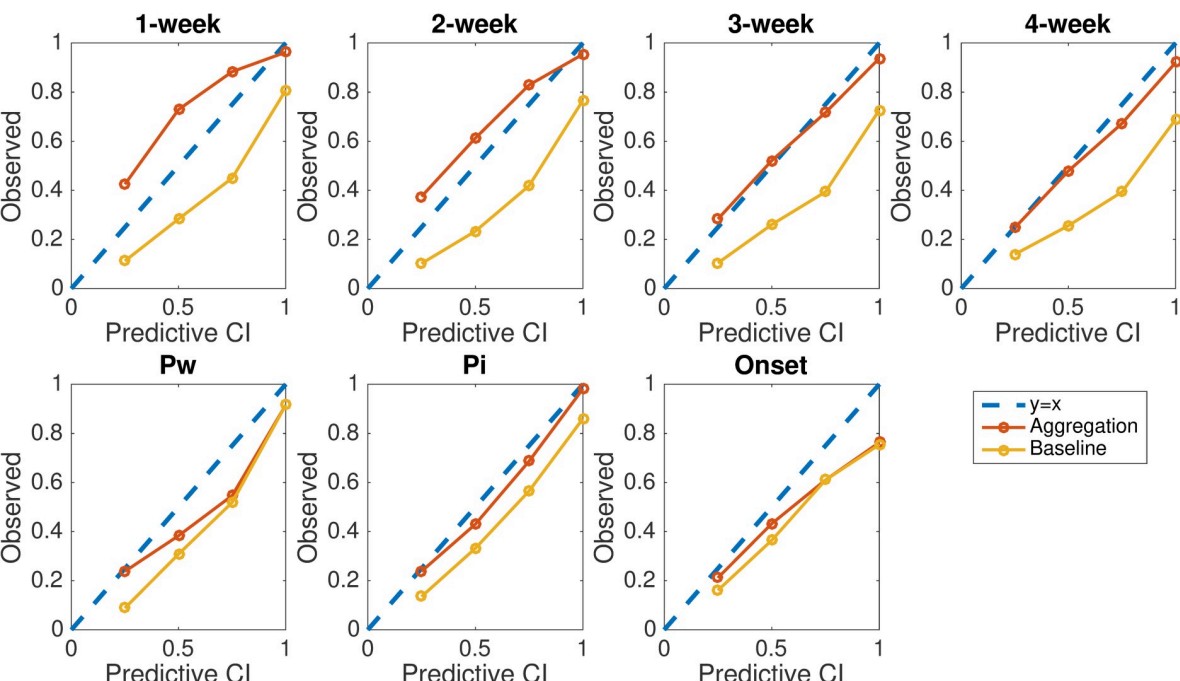

**Fig 5. Reliability plots for seven targets of ILI forecast targets.** Reliability plots for 1- to 4-week ahead ILI, peak week, peak intensity, and onset week are compared for multi-pathogen and single-pathogen forecasts. Data points show the fraction of observed targets falling within the 25%, 50%, 75% and 100% prediction intervals. Results are obtained from retrospective forecasts during 15 seasons at the US national level and for HHS regions 1 to 9.

implemented a version of the multi-pathogen forecasting in which the component forecasts for influenza were generated using the method of analogues. We found that using the method of analogues for influenza could also support improved forecasting of ILI over the baseline single-pathogen method (S19 Fig). However, due to the large variations of influenza activity across different seasons, the improvement using the method of analogues is less than that obtained using process-based models.

To examine the marginal benefits of including each pathogen, we performed an analysis in which each pathogen was omitted from the multi-pathogen forecasting, and quantified the degradation of forecast accuracy (S20 Fig). We found that removing any one of the six pathogens leads to a degradation of log score and omitting A/H3 leads to the largest degradation. This finding is in agreement with our estimated relative contribution to ILI reported in Table 1: A/H3 is the dominant signal in 9 of examined 15 seasons.

## Relative contribution of each pathogen

An additional benefit of the aggregation forecasting method is its inference and forecast of the relative contribution of each pathogen to ILI. Specifically, we computed the relative contribution of virus $i$ at week $t$ as $w_i v_i(t) / \sum_{i=1}^{6} w_i v_i(t)$, where $w_i$ is the multiplicative factor for virus $i$ estimated from available data. Here we only consider *the relative activity among the examined six viruses*. Should information about other ILI-related pathogens become available, it could be readily included into the framework. Fig 6 shows an example forecasting the relative contributions of the six viruses. Based on the seasonality of each pathogen, it is straightforward to reason that PIV12 and PIV3 would play dominant roles at the beginning and end of the respiratory virus season. However, the exact relative contribution depends on the evolving

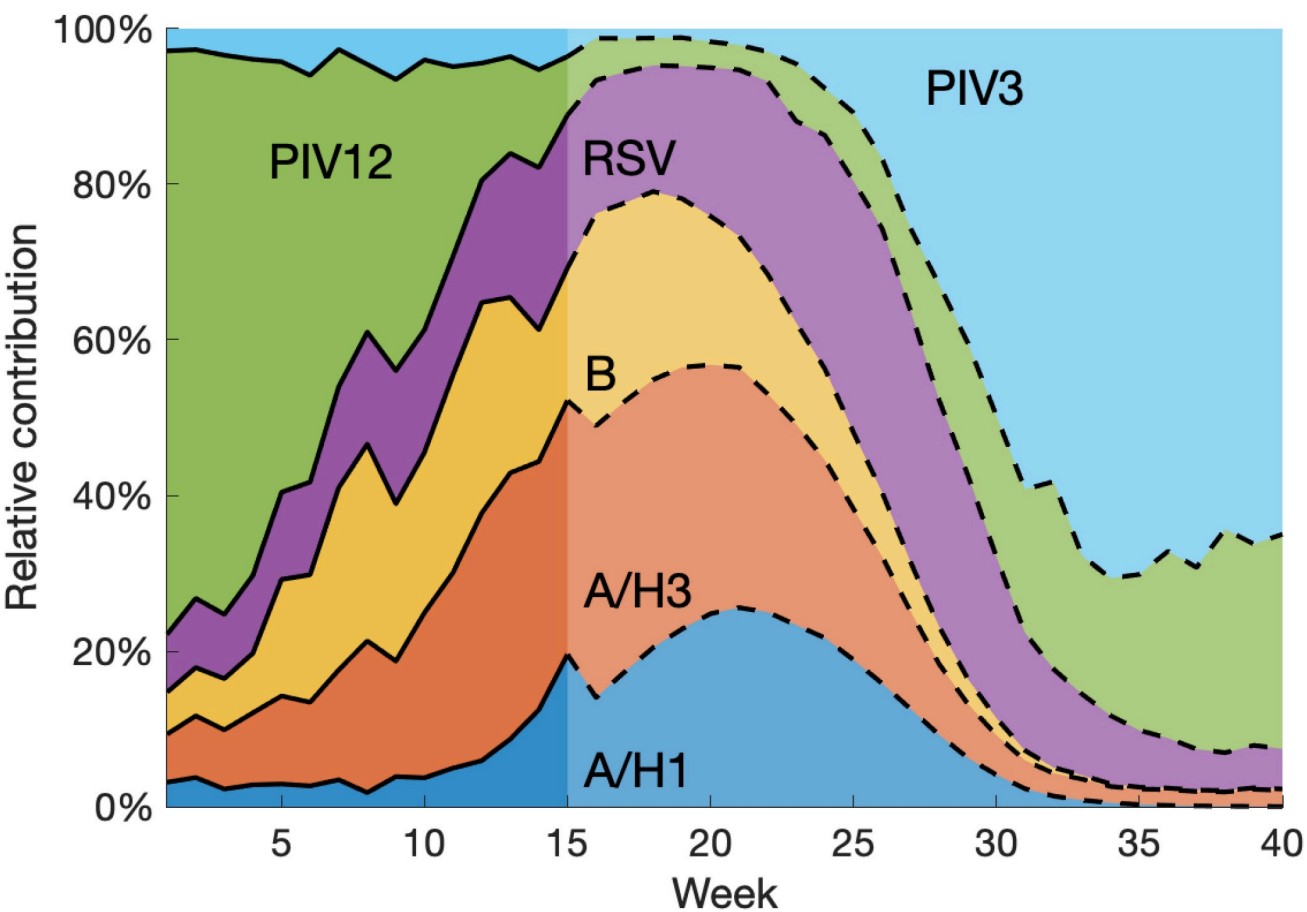

**Fig 6. Forecasting the relative contribution of each pathogen to ILI.** Forecasts were generated for national data at week 15 during the 2010–2011 season. The relative contribution before week 15 was inferred using available data. After week 15, it is obtained by normalizing the weighted predicted signal of each pathogen. Note that we only considered the relative contribution among the six examined viruses. Contributions from other pathogens were not included.

activities of all six pathogens. Reporting on the proportion of ILI attributable to different respiratory pathogens has large variations depending on cohort and study location [48–52]. For reference, please see Table 1 in Reis & Shaman [21]. The multi-pathogen forecast system, capable of generating accurate predictions for each virus, predicts the temporally varying relative contributions. In this example, during the period of peak ILI, influenza strains account for the majority of ILI attributed to the six examined viruses. This highlights the significant role influenza has in shaping ILI seasonality and explains the large variation of ILI curves among different seasons.

## Discussion

In this study, we demonstrate that a process-based model aggregating multiple ILI-causing pathogens can better describe overall ILI activity and support more accurate and precise forecasting. Our approach, established on the premise that ILI is the amalgamation of many infectious agents, proves particularly effective. In the 2019–2020 flu season, we have submitted the multi-pathogen ILI forecasts to the FluSight initiative. The performance is competitive, with potentials for further improvement. In addition to improved forecast accuracy for ILI, the system enables the estimation and forecast of the relative contribution of each virus to ILI. In

clinical settings, this information may be helpful for determining the possible cause of ILI among presenting patients.

Infectious disease forecasting is increasingly used to support public health decision-making [53,54] and have been particularly important during the ongoing COVID-19 pandemic. For intervention planning, forecasting the ILI signal may not generate sufficiently specific information. Even for typical wintertime cold and flu seasons, pathogen-specific forecasts could provide guidance on more specific intervention measures. For instance, in the 2019–2020 flu season, type B influenza appeared earlier and was abnormally high in several southern states (e.g., Louisiana) [2]. Anticipating an early surge of influenza B patients can support a better preparedness to this specific pathogen in healthcare systems.

Our findings indicate that influenza type A viruses are the predominant determinant of ILI seasonality. As a result, monitoring the activity of influenza type A viruses should be prioritized for accurate surveillance and forecasting of ILI. We further demonstrate that, through the inclusion of more realistic processes and virologic surveillance data, improved ILI forecasts can be generated to better support decision-making and intervention. Further improvement of process-based ILI forecasting will require additional representation of other information and processes (e.g., age groups, contact mixing and antigenic drift) and assimilation of more comprehensive datasets.

In the multi-pathogen model, only six pathogens were considered. This neglect of other pathogens may cause overestimation of the relative contributions of these viruses. In particular, the contributions of neglected pathogens are either included in the forecast discrepancy, or absorbed by viruses with similar seasonality. Considering this caveat, the contributions of PIV12 and PIV3 at the beginning and end of seasons are likely to be overestimated, possibly due to pathogens such as rhinovirus that circulate at the same time of year. Pathogens that peak in winter, e.g., coronavirus, may disturb the estimation during peak weeks. In addition, co-infection was not considered in the framework, which may also lead to model misspecification. However, co-infection of respiratory viruses is rare based on a recent study that actively tested 2,685 participants recruited at a New York City tourist attraction [55]. As data for more pathogens are made available, these issues may be corrected through incorporation into this forecasting framework.

For this study, information on test sensitivity/specificity was not available. Such details could be used to inform constraints of the observation error variance (OEV) of the positivity rate data. For instance, test results with high sensitivity and specificity would be assigned low OEV, which would give more credibility to these observations and impact the final estimated multiplicative factors. In operational forecasting, ILI data are subject to revisions for several weeks after the initial release. In this study, we did not consider this backfill; however, the impact of backfill should be limited as the aggregation procedure uses all available data points, while backfill mostly only affects the most recent 2 or 3 weekly data points. In practice, this backfill issue could be partly alleviated using nowcast techniques [56]. In future studies, the proposed method could also be used to disaggregate ILI or ILI-related hospitalizations by age groups, or to aggregate forecasts across different geographical scales (e.g., from state level to regional level).

## Supporting information

**S1 Text. Supplementary materials.**
(DOCX)

**S1 Table. Residuals of multi-pathogen and single-pathogen model fittings to ILI.** We measure the residual of model fitting to ILI using the sum of absolute discrepancies at all weeks in

each season and location. The first numbers are the residuals of multi-pathogen model fittings, and the latter ones are obtained from single-pathogen model fittings. The smaller fitting residuals are in bold. In general, the multi-pathogen model fittings have smaller residuals in most seasons and locations.
(DOCX)

**S1 Fig. Positivity rates of six pathogens in 15 seasons at national level.** We plot the positivity rates from laboratory tests at the national level for influenza A/H1, A/H3, B, RSV, PIV12 and PIV3 for the 1997–1998 to 2013–2014 seasons, excluding the 2008–2009 and 2009–2010 seasons. While influenza activities have large variations across seasons, non-influenza pathogens are more regular.
(EPS)

**S2 Fig. Scatter plot of each possible pair of observations.** We report the correlation coefficient of positivity rates between each pair of pathogens.
(EPS)

**S3 Fig. Residuals of multi-pathogen fit to ILI in the US and HHS regions 1 to 9.** Dash lines are residuals obtained in 15 seasons, and solid lines are the averaged curves. The mean residuals were used as the systematic bias that is the same across seasons in post-processing. Note that in most regions the decrease of ILI during the Christmas-New Year holiday is captured by the average residuals.
(EPS)

**S4 Fig. Estimated relative contributions of nine respiratory pathogens at the national level.** We estimated the relative contribution of each pathogen from 2010 to 2013, when positivity rate data from rAD, HMPV and Rhino were also available.
(EPS)

**S5 Fig. Comparison of multi-pathogen and single-pathogen model fittings to ILI.** Fitting was performed for national data for the 1997–1998 to 2013–2014 seasons, excluding the 2008–2009 and 2009–2010 seasons. The regression curves of multiple viral signals show less discrepancy from observations than the single-pathogen fitting.
(EPS)

**S6 Fig. Estimated distributions of multiplicative factors in different seasons at the national level.** Multiplicative factors vary considerably over time, indicating that contributions from the examined six viruses to ILI are variable in different seasons. The prior range for $w_i$ is set as $[0, 1]$.
(EPS)

**S7 Fig. Estimated multiplicative factors of six respiratory pathogens in the 2011–2012 season.** The prior range for $w_i$ is set as $[0, 1]$.
(EPS)

**S8 Fig. Estimated multiplicative factors of six respiratory pathogens in the 2013–2014 season.** The prior range for $w_i$ is set as $[0, 1]$.
(EPS)

**S9 Fig. Examples of forecasts for the six viruses.** Forecasts are shown for national positivity rates during the 2006–2007 season at week 10. The grey lines are mean forecast trajectories and grey areas represent the 95% predictive CIs.
(EPS)

**S10 Fig. Reliability plots for the forecasts of individual pathogens.** We display the reliability plots for 4 near-term targets: 1- to 4-week ahead ILI, denoted by X1 to X4, respectively. Data points show the fraction of observed targets falling within the 25%, 50%, 75% and 100% prediction intervals. For a well-calibrated forecast, data points will lie on the diagonal line y = x. Results are shown for the forecasts generated at the national level for all 15 seasons.
(EPS)

**S11 Fig. Examples of ILI forecasts at different phases of the 2010–2011 outbreak.** Forecasts are shown for national ILI during the 2010–2011 season at weeks 8, 13, 18 and 26. The grey lines are mean forecast trajectories and grey areas represent the 95% predictive CIs.
(EPS)

**S12 Fig. Examples of ILI forecasts at different phases of the 2011–2012 outbreak.** Forecasts are shown for national ILI during the 2010–2011 season at weeks 8, 13, 18 and 26. The grey lines are mean forecast trajectories and grey areas represent the 95% predictive CIs.
(EPS)

**S13 Fig. Examples of ILI forecasts at different phases of the 2012–2013 outbreak.** Forecasts are shown for national ILI during the 2010–2011 season at weeks 8, 13, 18 and 26. The grey lines are mean forecast trajectories and grey areas represent the 95% predictive CIs.
(EPS)

**S14 Fig. Examples of ILI forecasts at different phases of the 2013–2014 outbreak.** Forecasts are shown for national ILI during the 2010–2011 season at weeks 8, 13, 18 and 26. The grey lines are mean forecast trajectories and grey areas represent the 95% predictive CIs.
(EPS)

**S15 Fig. Comparison of log score and forecast error at different forecast weeks.** We compare the log score (left) and forecast error (right) for retrospective forecasts generated by aggregating predictions of multiple pathogens and a baseline method that simulates ILI as a single pathogen. Results are obtained by averaging the score or error for forecasts generated from week 4 to week 35 during 15 seasons at the US national level and for HHS regions 1 to 9.
(EPS)

**S16 Fig. Effect of post-processing on forecasts.** We compare the log scores (A) and forecast errors (B-C) for seven targets averaged over weekly predictions in 15 seasons and 10 locations (national and 9 HHS regions). Forecasts obtained by direct aggregation and additional postprocessing procedures (adjusting systematic bias, adjusting current bias and calibration) are compared with the baseline.
(EPS)

**S17 Fig. Comparison of the average width of the 95% prediction interval.** Results are obtained from retrospective forecasts during 15 seasons at the US national level and for HHS regions 1 to 9.
(EPS)

**S18 Fig. Effect of systematic bias correction on baseline forecast log score.** We compare the log scores for seven targets averaged over weekly predictions during 15 seasons and 10 locations (national and 9 HHS regions). Forecasts obtained using the baseline with systematic bias correction (Baseline+systematic bias correction) are compared with the baseline forecast method (Baseline).
(EPS)

**S19 Fig. Forecasting influenza using the method of analogues in multi-pathogen forecasting.** We compare the log scores for seven targets averaged over weekly predictions in 15 seasons and 10 locations (national and 9 HHS regions). Forecasts were obtained using the multi-pathogen method in which influenza types/subtypes were predicted using dynamical models (Dynamic), the multi-pathogen method in which influenza types/subtypes were predicted using the method of analogues (Analogues) and the baseline single-pathogen method (Baseline).
(EPS)

**S20 Fig. Degradations of forecasts if each pathogen is removed from the multi-pathogen forecasting system.** We compare the log scores for seven targets averaged over weekly predictions in 15 seasons and 10 locations (national and 9 HHS regions). We removed each pathogen in turn in the multi-pathogen forecasting system and compared the degradation of log score.
(EPS)

## Author Contributions

**Conceptualization:** Sen Pei, Jeffrey Shaman.

**Data curation:** Sen Pei.

**Formal analysis:** Sen Pei.

**Funding acquisition:** Jeffrey Shaman.

**Investigation:** Sen Pei, Jeffrey Shaman.

**Methodology:** Sen Pei, Jeffrey Shaman.

**Supervision:** Jeffrey Shaman.

**Validation:** Sen Pei.

**Visualization:** Sen Pei.

**Writing – original draft:** Sen Pei.

**Writing – review & editing:** Jeffrey Shaman.

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
