## [Decision Letter · Decision Letter 0]

4 Mar 2020

Dear Dr. Pei,

Thank you very much for submitting your manuscript "Aggregating forecasts of multiple respiratory pathogens supports more accurate forecasting of influenza-like-illness" for consideration at PLOS Computational Biology.

As with all papers reviewed by the journal, your manuscript was reviewed by members of the editorial board and by several independent reviewers. In light of the reviews (below this email), we would like to invite the resubmission of a significantly-revised version that takes into account the reviewers' comments. Note that the reviewers appreciated the attention to an important problem, but raised some substantial concerns about the manuscript as it currently stands. While your manuscript cannot be accepted in its present form, we are willing to consider a revised version in which issues raised by the reviewers have been adequately addressed. We cannot, of course, promise publication at that time.

Sincerely,

Sara Y Del Valle

Guest Editor

PLOS Computational Biology

Rob De Boer

Deputy Editor

PLOS Computational Biology

Reviewer's Responses to Questions

**Comments to the Authors:**

Reviewer #1: Review of “Aggregating forecasts of multiple respiratory pathogens supports more accurate forecasting of influenza-like-illness”

The authors present a well-motivated paper. ILI forecasting has garnered significant public health attention in recent years. ILI, though often treated like a singular entity, is really a collection of numerous respiratory illnesses, including the flu, as the authors state. Moving ILI forecasting models in directions that better align with the ILI data-generating process represents an important step for ILI forecasting. Furthermore, the authors present evidence that forecasting can be improved by such a move.

In principle, I’m bullish on this work. That said, the execution left much to be desired. Numerous issues exist and clarifying questions need to be addressed. By far the most significant issue surrounds the prior specification and sampling of the weights, w_i.

All issues and clarification questions are enumerated below.

Major issues:

1. Do the weights w_i described in lines 166-171 as well as the “Aggregation via MCMC” section of the Supplementary Materials (SM) sum to 1? The authors never say that they do, but they do describe ILI as being the aggregation of 6 pathogens, suggesting that the weights should sum to 1. It seems like they should but not clear if they do. Clarification is needed.

2. How are the weights actually estimated? On lines 175-176, the authors say, “We then performed a linear regression to ILI using the fitted curves via MCMC in order to derive estimates for w_i (Fig 1B).” The authors, in the second paragraph of section “Aggregation via MCMC” of the SM go on to say, “The prior distribution for each weighting factor was set as a uniform distribution between 0 and 0.3. Starting from a set of weighting factors randomly drawn from the prior distributions, a Metropolis algorithm was applied. At each update, a new set of weighting factors was obtained by sequentially perturbing w_i.” Because a Uniform [0, 0.3] prior was selected for each weight, that means the posterior must necessarily be between 0 and 0.3, inclusively. That check seems to hold for Figure S5, as the posterior distribution for each weight and each season appears to be between 0 and 0.3. That same check, however, does not hold for the weights in Fig 1B. Specifically, some of the posterior draws for the PIV12 weight exceeds 0.3, which is not possible if they were given a prior between 0 and 0.3. Explanation is needed. Shouldn’t Fig 1B be identical to the 2010 panel of Fig S5? Relatedly, what proposal density is used for the Metropolis algorithm?

3. It’s hard to understand what value the posteriors of the weights have, given that they were assigned a seemingly arbitrary prior hard capped above at 0.3 and many of the posterior distributions are bumping up against this arbitrary upper bound (e.g., Fig S5 - PIV12 in 2001, 2002, 2010, and 2011). This suggests that the data actually wants the weight for PIV12 to be larger than 0.3 but it can’t because of the selected prior. In general, there is nothing wrong when a posterior bumps up against a prior upper or lower bound when that upper or lower bound represents a physical constraint. When a prior bumps up against an arbitrary bound, however, that is a clear sign of prior misspecification. In such a case, a new prior should be selected that pushing those bounds out as the fits and likely the forecasts are suffering from this arbitrary 0.3 upper bound choice. It seems a more physical and elegant solution would be to assign a Dirichlet prior to the weight vector (all 6 weights jointly). A Dirichlet prior (and Metropolis-Hastings proposal) will guarantee that 1) all weights are positive and 2) all weights will sum to 1. Thus, if a weight bumps up against its lower (0) or upper (1) bound, that will be alright because those bounds represent physical lower and upper weight limits. If the authors choose not to change their priors and keep the analysis as is, they will need to provide extremely strong arguments as to why the weights cannot exceed 0.3.

Issues:

1. Throughout the manuscript, the authors say their models are “accurate” in an absolute sense (e.g., line 178; lines 218-220). However, the authors never define what “accurate” means in an absolute sense leaving the reader to define this for themselves. This is challenging to do as some of the language the author’s use deviates from my interpretations. To me, for instance, the red line in the Strain B panel of Fig 1A deviates from the observations roughly as significantly as the yellow line (single agent) does in Fig 1C. However, the authors say, “The mechanistic models accurately reproduced the positivity rates of each pathogen (Fig 1A)” (line 178) to describe the Strain B fit while simultaneously describe the single pathogen fit in Fig 1C as having, “large discrepancies from observations” (line 183). If the authors choose to describe their fits as “accurate” in an absolute sense, they need to provide an unambiguous definition of what “accurate” means and then demonstrate their fits either do or do not meet that unambiguous definition. Otherwise, the authors should refrain from such absolute claims.

2. Table 1: The percentages within a row in Table 1 do not sum to 100%. Are they supposed to? It seems like they should. I don’t know how to interpret them if they don’t.

3. Lines 230 and 240: Please provide more detail regarding how the Wilcoxon signed-rank test was performed. Its appropriateness is unclear based on the information provided.

4. Line 257: A conclusion of this work is that for the 2010-2011 season, flu strains account for up to 80% of ILI of the six examined viruses. How does this 80% estimate compare to the author’s previous work with ILI+, which is a scaling of ILI by the proportion of ILI patients testing positive for influenza. At first blush, 80% seems high.

5. SM second equation: The fraction should be flipped, with P(T|M) in the denominator. This means the fraction in the third equation also needs to be flipped and the interpretation of the weight is backwards. In general, the “Disaggregation of ILI” section should be closely examined and updated in light of the error in the second equation.

6. Table S1: It’s not intuitive to me why the multi-pathogen fit should ever be worse than the single-pathogen fit. Isn’t the model space of the single-pathogen contained within the more flexible multi-pathogen model space? Said another way, the multi-pathogen fit can take any form the singe-pathogen fit can take, but the single-pathogen fit cannot take any form the multi-pathogen fit can take. If this is true, it’s unclear why, when fitting (not forecasting) the multi-pathogen fit should ever be worse than the single-pathogen fit.

7. Figure S10: The central thesis of the paper is that forecasting ILI as a weighted sum of its constituent pathogens yields better forecasts than forecasting ILI directly as a single pathogen. This conclusion is supported by Fig S10. The authors also demonstrate that forecasting can be further improved by accounting for systematic bias. The glaring hole in the comparison in Fig S10 is the forecasts of the single-pathogen fit (baseline) that accounts for systematic bias. Without this comparison, it’s hard to know which modeling technique provides the largest improvement: the multi-pathogen modeling (as is suggested by the thesis of the paper) or the systematic bias modeling. The single-pathogen with systematic bias would be a very welcomed contribution to the paper.

Minor issues:

1. ILI is “influenza-like illness” not “influenza-like-illness” (e.g., the title and the abstract)

2. Line 109: In what sense is the perturbing “optimal”? Please provide a bit more context.

3. Line 110: please use consistent formatting for references (e.g., [28] instead of superscripting).

4. Lines 162-163: Why are RSV, PIV12, and PIV3 more regular than flu strains? Why do they peak when they do? Is this known? In general, a bit more context should be added to RSV, PIV12, and PIV3 – both what we know and don’t know about their seasonality.

5. Lines 166-168: Is there a season index being suppressed in the ILI(t) equation? For instance, is this ILI(t) for season s?

6. Line 172: It’s not clear how the dependent clause, “To eliminate the impact of observational error,” is related to the rest of this sentence. How is the fitting of each respiratory pathogen related to the observation error?

7. ILI data is subject to revisions for many weeks after the initial release of the ILI data. Is backfill accounted for in this work? If not, why not and how do you imagine backfill would impact this work?

8. SM last equation in section “Forecasting Influenza”: If you sample a negative positivity rate, do you set it equal to zero (as you know a positivity rate can’t be negative) or do you leave it negative?

9. Fig S7 and S11: The 95% prediction intervals are plotted at the x=1 point, not the x=.95 point. Is what’s plotted a 95% prediction interval at the wrong point, or a 100% prediction interval at the right point?

10. Table S1: Should “sum of discrepancies” in the Table caption be “sum of squared discrepancies” or “sum of absolute discrepancies”? All reported figures are non-negative, suggesting the sums are not the raw discrepancies.

Reviewer #2: This manuscript builds on a series of prior work from these authors to improve the accuracy of influenza-like illness modelling in the US. It is motivated by the flusight forecasting challenge. In addition to improved forecasting, these studies are fundamentally moving the needle in our epidemiological understanding of influenza and (in this case) of respiratory pathogens in general.

In this manuscript, the authors draw on type-specific regional data to improve their forecasts of national and regional ILI. They use their existing mechanistic models for influenza subtypes and then the method of analogues for the non-flu pathogens. They find consistently increased accuracy of retrospective forecasts with their multi-pathogen approach.

I enjoyed reading the paper and I think it’s a substantial contribution. Forecasting aside, we have very few disease-dynamic studies of multiple pathogens over multiple years. The lower variance of incidence for the other types versus influenza is in itself an interesting result. The authors may wish to comment on the type of immunity and transmissibility of influenza versus the other pathogens. I wonder if influenza has a lowish R0 and long leaky immunity that leads to the variation versus the others?

I have a number of comments around identifying the marginal contribution of different data and approaches to the increased accuracy of the forecasts. However, although I might have made a few different design choices, the merit of the retrospective forecast design (against an established baseline) is that that mis-specification and over-fitting are far less of a worry than a typical likelihood-driven study.

Did the authors explicitly try the method of analogues for the influenza dynamics? I appreciate that they have a lot of experience with the the process models, but given that the toolkit must be to hand in this project, it would be good to know that the process models do do better than the method of analogues and by how much on the metrics used here. Constant scepticism about the merit of prcioess models will result in really good process models at some point!

How much did each extra pathogen improve the accuracy? The authors state that more pathogens might be even better, but I wonder if the improvement in accuracy is coming from just a few of the additional non-flus. It would be interesting (and hopefully only resource intensive in computer time rather than human time) to hold each pathogen out in turn and assess the degradation of the forecast.

Many of these data will come from multiplex PCR…, but they are being treated as independent observations that contribute to ILI. Are there additional published data from multiplex studies that directly measure the proportion of ILI attributable to each type? Would sample-linked data be able to confirm any of the underlying model parameters. Are there estimates out there for the proportion of ILI in the US due to these 6?

The charts, both main text and SI are very clear and allow the reader to understand the work quickly.

These authors are part of the ongoing effort with the flusight initiative. Have they used these methods in the competition yet? Might be worth a comment in the discussion without going into too much detail.

Some specific comments

Line 110; there has been a recent revision in the flusight scoring algorithm, discussed in an exchange of letters in PNAS. https://doi.org/10.1073/pnas.1912147116 Does this paper use the old or the new algorithm. Either is fine, but please state and cite the exchange.

S1 Figure. Its a little difficult to spot patterns here and these are very innovative data. I wonder if scatter plots of each possible pair of observations (so time removed) might give a sense for the amplitude of different pathogens, their variance and their correlation with other pathogens.

S2 Figure. I’d put this in the main text. These are the outcome and “exposures” for this study and the patterns here are pretty clear.

144; I am more than happy to work in the log score, but maybe include a bit of narrative for the reader about going from score to skill. … -0.5 is correct approx 61% of the time !!

172; to eliminate the impact of observational error - I couldn’t quite understand this.

178; see comments above, would it be worth testing analogues for influenza as well as the non-flu viruses

194; B looks more like the non-flus in this regard?

237; % improvement in log scores is a but difficult to interpret. Maybe state the change in log scores or give % changes in skill rather than score?

253; Are there other sources of evidence about proportions of ILI that arise from different pathogens? See comments above.

276; If arguing for more pathogens being better, would be good to check the incremental impact of each of the pathogens here. See other comments above.

The SI was very clear. I didn’t see the SI data (and Imight have had a play with these!) but assume that it will be uploaded on publication.

Reviewer #3: The main claims of the paper are that modeling ILI as a weighted average of its contributing viral signals leads to more accurate and precise ILI forecasts (relative to a single-pathogen model for ILI), and that such disaggregation also has value from a public health perspective. The originality and innovative nature of this paper rely largely on the decomposition of ILI into component pathogenic contributors. However, it is not entirely clear based on the current intro whether/how such decomposition has been done before. Lines 31-32: "most existing process-based forecasting systems treat ILI as a single infectious agent." Most, but not all? Can you cite some forecasting systems that do not treat ILI as a single infectious agent? How do these differ from your method? (Same in line 62, with the language "almost all".) If there are competitors that also decompose ILI into component pathogens, the specific contributions of this paper relative to those competitors needs to be made explicit.

As far as the claim of improved accuracy and precision, the former is thoroughly demonstrated relative to the single-pathogen SIR model and the latter is not rigorously shown. We can see visually in Fig 2 that the 95% CI of the baseline model seems wider for that particular forecast example, but a table or figure giving a comparison of the area of the models' 95% predictive CIs is needed to make this a general claim.

Performance is compared to what seems to be to be a 'bad' baseline that simulates ILI as a single pathogen. How does performance at predicting overall ILI compare to other competitors that don't consider separate pathogens (e.g., the purely statistical top-performing FluSight model Dante)? Obviously the authors' method has advantages from an interpretability standpoint, but if it performs more poorly than a purely statistical model this would imply that the process-based model is still not quite capturing reality very well. Such a result would not invalidate the claims of the paper, or render its contribution irrelevant, but would indicate that further understanding of the process driving the ILI curve is needed.

The authors argue that learning about the trajectories of the individual infectious agents is potentially useful in clinical settings for determining the possible cause of ILI among multiple infectious agents. This claim of pathogen-specific forecasts improving on public health response relative to ILI forecasts alone could be strengthened by a citation. Additionally, the relationship between the sampling protocol for ILI and that of virologic surveillance isn't made entirely clear in the paper. How are virologic surveillance data sampled? Does this sampling protocol vary by year, and does it impact predictions? I'd like to know more about potential biases in the virologic data, and what the underlying population is of the positivity rate (Is it ILI patients? Is it people at the hospital? Are sicker patients more likely to get a virologic test? Etc.).

I am concerned about the assumption that ILI is a weighted sum of the 6 included pathogens. I think that slightly more care needs to be paid to this issue in the main text outside of the discussion section. I'd like to know whether it would be possible in this framework to include a catch-all "other" category. Given that the authors make the argument that accounting for the various pathogen sources of ILI activity has advantages from both a predictability and an interpretability perspective, it seems misguided for the model itself to implicitly not acknowledge that ILI has other contributors than the 6 pathogens having historic data.

In addition to discussing how (or why not) to include an 'other' category, I think more justification beyond data availability could ameliorate some of the concerns of including only those 6 pathogens. Specifically, I want to know what proportion of ILI seems to not be covered by those pathogens having data since 1997. That is, for the years that data are available, what are the positivity rates for the example other-pathogens cited on lines 93-94 (coronavirus, human metapneumovirus and respiratory adenovirus)? If the positivity rates are high, I would worry more that the 6 included pathogens aren't actually what comprises much of ILI. If low, I'm less concerned.

In line 167-168, the authors should be clear about whether or not \\sum w_i = 1. Does the weighting factor imply anything about whether some of ILI can be driven by an individual having multiple pathogens? Is this assumption valid? You mention this briefly in the discussion, but a quick note here would also be helpful. I'm also interested in the relationship between the weighting factor and the sensitivity/specificity of the test. If you knew about the TPR/FPR/FNR of the test could this model incorporate that information?

Figure S5 shows the estimated distributions of weighting factors in different seasons at the national level. It would be useful to also include a similar figure (or a few figures) with season fixed and weighting by region. I'm curious if there is significant regional variability.

Also in looking at Figure S5 I had the question as to why weighting factors should vary so considerably by season. The authors mention that this is due to contributions themselves being variable. However, it would make more sense to me for the weights to be consistent across seasons, but the positivity rate itself to drive differing pathogenic contributions. The variability of the weighting factors seems to me to perhaps be an artifact of sampling differences across seasons in the virologic data. I think this needs further discussion/consideration.

I would also like some comment/discussion from the authors on why the weights for some pathogens in Figure S5 have such small spread (e.g., RSV) while others are hugely variable (e.g., many have box-whisker plots spanning from near-0 to above 0.2). Is there a set of good solutions to the data that can come from essentially 0-weighting a pathogen, while putting all the weight on a different one? Is the set of good solutions highly modal, or is there a smooth trajectory of weights that yield a good fit?

A handful of smaller comments:

- Figure 4 is beautiful!!

- Is Figure S6 a remarkable example (particularly good or bad)? Having just one such example makes me wonder why it was chosen, so perhaps include a few more and note the performance relative to overall average performance.

- In line 109-110 what is the superscript 28 referring to?

- I appreciate the authors noting that the particular structure of the model (modeling ILI as an aggregation of component parts) isn't unique to pathogens but could be used to aggregate forecasts by age group or geography.

- Is the code available? For reproducibility it would be helpful to make this public.

**Have all data underlying the figures and results presented in the manuscript been provided?**

Reviewer #1: Yes

Reviewer #2: No: The data are stated as being available but were not present.

Reviewer #3: Yes

PLOS authors have the option to publish the peer review history of their article (what does this mean?). If published, this will include your full peer review and any attached files.

Reviewer #1: No

Reviewer #2: No

Reviewer #3: No
---

## [Decision Letter · Decision Letter 1]

30 Jul 2020

Dear Dr. Pei,

Thank you very much for submitting your manuscript "Aggregating forecasts of multiple respiratory pathogens supports more accurate forecasting of influenza-like illness" for consideration at PLOS Computational Biology. As with all papers reviewed by the journal, your manuscript was reviewed by members of the editorial board and by several independent reviewers. The reviewers appreciated the attention to an important topic. Based on the reviews, we are likely to accept this manuscript for publication, providing that you modify the manuscript according to the review recommendations.

Sincerely,

Sara Y Del Valle

Guest Editor

PLOS Computational Biology

Rob De Boer

Deputy Editor

PLOS Computational Biology

[LINK]

Reviewer's Responses to Questions

**Comments to the Authors:**

Reviewer #3: I thank the authors for their thorough and thoughtful responses to the reviewers! In light of the clarifications/additions, I feel that this manuscript is ready for publication.

I had one small note-- In one of the responses the authors note: “Dante is not a purely statistical model. It couples a process-based SIR model with a statistical procedure to correct forecast discrepancies.” I believe that they are referring to DBM (https://arxiv.org/abs/1708.09481, also by Osthus). Unlike DBM, Dante (https://arxiv.org/abs/1909.13766) does not rely on a process-based SIR model and is in fact a purely statistical model. In spite of this issue, the authors' response still addressed the underlying question, so thank you for noting the relative performance of your model to Dante.

Reviewer #4: The authors have submitted a revised manuscript describing their work on aggregating multiple pathogens to improve ILI forecasting. The initial reviews were generally positive, mostly focusing on clarification and contextualization of the authors' work. The authors have addressed the first round of reviews, and the clarity of the paper is noticeably improved. I do not have any major comments or criticism of the work as is. I think it should be published because it demonstrates a novel ILI forecasting strategy that has the potential for future improvements. I think that the paper's results may be challenging to reproduce verbatim from the supplementary material. Still, because the Authors have said they will post their code to GitHub, I am not concerned. I offer a handful of minor comments.

Comments:

line 68-69: Could you insert a quick definition of "positivity rate" here. Just for completeness.

line 122: Could "relatively regular outbreaks" be changed to something like "consistent seasonality"? The term regular outbreaks I don't think describes the temporal pattern you are focusing on. Regular outbreaks could mean than they happen annually, whereas you are specifically keying on the regularity of the outbreak time within any given year.

line 133: Sampled with replacement?

line 140-141: Explicitly state that the prior is uniform(0,1) instead of just describing a range

line 145: The word "inferred" is redundant because you infer the true posterior by performing the MCMC sampling

line 355: In this sentence, you say that "forecasting an aggregated ILI signal is insufficient," but forecasting ILI using an aggregation model is what this paper is about. I think you can drop the "aggregated" to emphasize the point that you are trying to make in this paragraph, that is, modeling separate pathogens provides more information and solutions to decision-makers.

line 636: The units of the "viral signals" in Figure 1 is not explicit. I am assuming it is the observed positivity rate.

SI Eq. 5: The sign of the I/D term is wrong. It should be negative.

SI pg. 2: The second paragraph is unclear. What does "free simulation" mean in this context?

SI pg. 4: The assumption of the distribution for $\\bar{v}_i(t)$ seems to be implying a Bayesian posterior using an improper prior. Is there a reason for not being explicit about this choice of inference?

SI pg. 4: In paragraph 3, why was the ensemble of forecast trajectories for each pathogen not incorporated into the likelihood when performing the MCMC to sample the multiplicative factors? It seems like you accept/reject the multiplicative factors with one set of $v_i(t)$ and then switching to another set of $v_i(t)$ when performing the forecasting.

**Have all data underlying the figures and results presented in the manuscript been provided?**

Reviewer #3: Yes

Reviewer #4: None

PLOS authors have the option to publish the peer review history of their article (what does this mean?). If published, this will include your full peer review and any attached files.

Reviewer #3: No

Reviewer #4: No
---

## [Editor Report · Decision Letter 2]

2 Sep 2020

Dear Dr. Pei,

We are pleased to inform you that your manuscript 'Aggregating forecasts of multiple respiratory pathogens supports more accurate forecasting of influenza-like illness' has been provisionally accepted for publication in PLOS Computational Biology.

Best regards,

Sara Y Del Valle

Guest Editor

PLOS Computational Biology

Rob De Boer

Deputy Editor

PLOS Computational Biology

---

## [Editor Report · Acceptance letter]

14 Oct 2020

PCOMPBIOL-D-19-01318R2 

Aggregating forecasts of multiple respiratory pathogens supports more accurate forecasting of influenza-like illness

Dear Dr Pei,

I am pleased to inform you that your manuscript has been formally accepted for publication in PLOS Computational Biology. Your manuscript is now with our production department and you will be notified of the publication date in due course.

With kind regards,

Matt Lyles
